# Concept Activation Regions: A Generalized Framework For Concept-Based Explanations

**Jonathan Crabbé**
University of Cambridge
jc2133@cam.ac.uk

**Mihaela van der Schaar**
University of Cambridge
The Alan Turing Institute
UCLA
mv472@cam.ac.uk

## Abstract

Concept-based explanations permit to understand the predictions of a deep neural network (DNN) through the lens of concepts specified by users. Existing methods assume that the examples illustrating a concept are mapped in a fixed direction of the DNN's latent space. When this holds true, the concept can be represented by a concept activation vector (CAV) pointing in that direction. In this work, we propose to relax this assumption by allowing concept examples to be scattered across different clusters in the DNN's latent space. Each concept is then represented by a region of the DNN's latent space that includes these clusters and that we call concept activation region (CAR). To formalize this idea, we introduce an extension of the CAV formalism that is based on the kernel trick and support vector classifiers. This CAR formalism yields global concept-based explanations and local concept-based feature importance. We prove that CAR explanations built with radial kernels are invariant under latent space isometries. In this way, CAR assigns the same explanations to latent spaces that have the same geometry. We further demonstrate empirically that CARs offer (1) more accurate descriptions of how concepts are scattered in the DNN's latent space; (2) global explanations that are closer to human concept annotations and (3) concept-based feature importance that meaningfully relate concepts with each other. Finally, we use CARs to show that DNNs can autonomously rediscover known scientific concepts, such as the prostate cancer grading system.

## 1 Introduction

Deep learning models are both useful and challenging. Their utility is reflected in their increasing contributions to sophisticated tasks such as natural language processing [1, 2], computer vision [3, 4] and scientific discovery [5–8]. Their challenging nature can be attributed to their inherent complexity. State of the art deep models typically contain millions to billions parameters and, hence, appear as *black-boxes* to human users. The opacity of black-box models make it difficult to: anticipate how models will perform at deployment [9]; reliably distil knowledge from the models [10] and earn the trust of stakeholders in high-stakes domains [11–13]. With the aim of increasing the transparency of black-box models, the field of explainable AI (XAI) developed [14–17]. We can broadly divide XAI methods in 2 categories: ① Methods that restrict the model's architecture to enable explanations. Examples include attention models that motivate their predictions by highlighting features they pay attention to [18] and prototype-based models that motivate their predictions by highlighting relevant examples from their training set [19]. ② *Post-hoc* methods that can be used in a plug-in fashion to provide explanation for a pre-trained model. Examples include *feature importance methods* (also known as feature attribution or saliency methods) that highlight features the model is sensitive to [20–26]; *example importance methods* that identify influential training examples [27–29] and *hybrid*

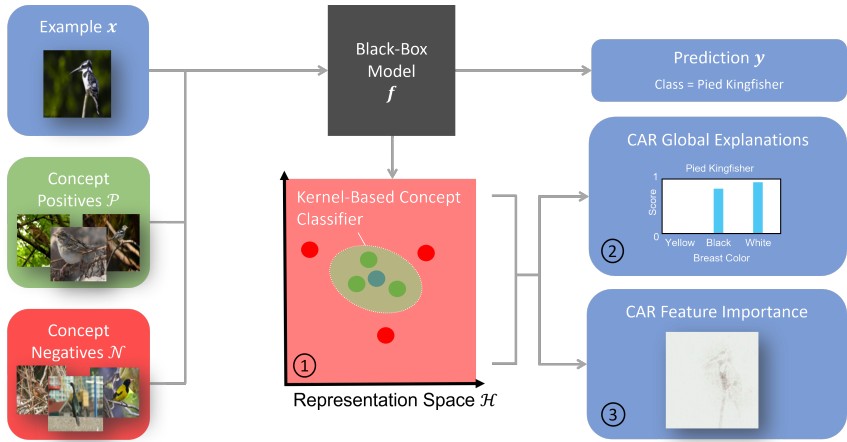

Figure 1: Illustration of the CAR framework. To define a concept, a set of positive and negative examples are fed to the model. In this illustration, we consider a neural network predicting the species of a bird based on a picture. The concept we illustrate is *white breast*. Concept positive and negative images respectively exhibit birds with and without white breasts. CARs aim at explaining the prediction for a given example $x$. ① All the examples are mapped in the model's representation space $\mathcal{H}$. Unlike CAVs [31], CARs rely on kernel-based support vector classifiers (SVC) and do not require the positive and negative sets to be linearly separable in $\mathcal{H}$. ② CAR uses the SVC to output TCAR scores that indicate how classes are related to concepts. ③ The CAR formalism can be used in combination with any feature importance method for deep neural networks. This permits to create concept-specific saliency maps.

*methods* combining the two previous approaches [30]. In this work, we focus on a different type of explanation methods known as *concept-based* explanations. Let us now summarize the relevant literature to contextualize our own contribution.

**Related work.** Concept-based explanations were first formalized with the *concept activation vector* (CAV) formalism [31, 32]. Given a concept specified by the user (e.g. stripes in an image), linear classifiers are used as probes [33] to assess whether a deep model's representation space separates examples where a concept is present (*concept positive examples*) from examples where a concept is absent (*concept negative examples*). A CAV is then extracted from the linear classifier's weights. With this CAV, it is possible to provide post-hoc explanations such as the sensitivity of a model's prediction to the presence/absence of a concept. It goes without saying that several concepts are needed to explain the prediction of a deep model. To formalize this idea, existing works have used concept basis decomposition [34] and sufficient statistics [35]. Although most applications of CAV involve image data, we note that the formalism has been successfully applied to time-series data [36, 37]. While concepts are typically specified by the user, early works in computer vision have been undertaken to discover concepts in the form of meaningful image segmentations [38]. The main criticism against the CAV formalism is that it requires concept positive examples to be linearly separable from concept negative examples [39, 40]. This is because linear separability of concept sets is a restrictive criterion that the model is not explicitly trained to fulfil. To address this issue, works like *concept whitening transformations* [39] and *concept bottleneck models* [41–43] propose to do away with the post-hoc nature of concept-based explanations. These methods introduce new neural network architectures that permit to train the models with concept labels. We stress that this requires the set of concepts to be specified *before* training the model. This assumes that we know what concepts are relevant for a model to solve a downstream task *a-priori*. This is not the case whenever we train a model to solve a task for which little or no knowledge is available. In this setup, it seems more appropriate to train a model for the task first and, then, perform a post-hoc analysis of the model to determine the concepts that were relevant in providing a solution. Furthermore, recent concerns have emerged regarding the reliability of interpretations provided by these altered model architectures [44, 45].

**Contributions.** In this work, our purpose is to retain the flexible post-hoc nature of concept-based explanations without assuming that the concept sets are linearly separable. To that aim, we introduce *concept activation regions* (CARs), an extension of the CAV framework illustrated in Figure 1.

① **Generalized formalism.** As a substitute to linear separability, we propose in Section 2.1 to adapt the smoothness assumption from semi-supervised learning [46]. Intuitively, this more general assumption only requires positive and negative examples to be scattered across distinct clusters in the model's representation space. In practice, this generalization is implemented by substituting CAV's linear classifiers by kernel-based support vector classifiers. We demonstrate that choosing radial kernels leads to CAR classifiers that are invariant under isometries of the latent space. This permits to assign identical explanations to latent spaces characterized by the same geometry. Moreover, we show in Section 3.1.1 that our CAR classifiers yield a substantially more accurate description of how concepts are distributed in deep model's representation spaces. ② **Better global explanations.** With the nonlinear decision boundaries of our support vector classifiers, there is no obvious way to adapt the notion of concept activation vectors. Since CAV's concept importance (TCAV score) is computed with these vectors, we need an alternative approach. In Section 2.2, we propose to define concept importance by building on our smoothness assumption. Concretely, a concept is important for a given example if the model's representation for this example lies in a cluster of concept positive representations. With this characterization, we define TCAR scores that are the CAR equivalent of TCAV scores. In Section 3.1.2, we demonstrate that TCAR scores lead to global explanations that are more consistent with concept annotations provided by humans. ③ **Concept-based feature importance.** In Section 2.3, we argue that our CAR formalism permits to assign concept-specific feature importance scores for each example fed to the neural network. We verify empirically in Section 3.1.3 that those feature importance scores reflect meaningful concept associations. Finally, we illustrate in Section 3.2 how these contributions permit to establish that deep models implicitly discover known scientific concepts.

## 2 Concept Activation Regions (CARs)

### 2.1 Preliminaries

We assume a typical supervised setting where each sample is represented by a couple $(\boldsymbol{x}, \boldsymbol{y})$ with input features $\boldsymbol{x} \in \mathcal{X} \subseteq \mathbb{R}^{d_X}$ and a label $\boldsymbol{y} \in \mathcal{Y} \subseteq \mathbb{R}^{d_Y}$, where $d_X, d_Y \in \mathbb{N}^*$ are respectively the dimensions of the feature (or input) and label (or output) spaces [1]. In order to predict the labels from the features, we are given a deep neural network (DNN) $\boldsymbol{f} = \boldsymbol{l} \circ \boldsymbol{g} : \mathcal{X} \to \mathcal{Y}$, where $\boldsymbol{g} : \mathcal{X} \to \mathcal{H}$ is a feature extractor that maps features $\boldsymbol{x} \in \mathcal{X}$ to latent representations $\boldsymbol{h} = \boldsymbol{g}(\boldsymbol{x}) \in \mathcal{H} \subseteq \mathbb{R}^{d_H}$ and $\boldsymbol{l} : \mathcal{H} \to \mathcal{Y}$ maps latent representations $\boldsymbol{h} \in \mathcal{H}$ to labels $\boldsymbol{y} = \boldsymbol{l}(\boldsymbol{h}) \in \mathcal{Y}$. The representation $\boldsymbol{h} = \boldsymbol{g}(\boldsymbol{x})$ typically corresponds to the output from one of the DNN's hidden layer. We assume that the model $\boldsymbol{f}$ was obtained by fitting a training set $\mathcal{D}_{\text{train}} \subset \mathcal{X} \times \mathcal{Y}$. Our purpose is to *understand* how the neural network representation induced by $\boldsymbol{g}$ allows the model $\boldsymbol{f}$ to solve the prediction task. To that end, we shall use concept-based explanations. Those explanations rely on a set of $C \in \mathbb{N}^*$ concepts indexed [2] by $c \in [C]$, where $[C]$ denotes the set of positive integers between 1 and $C$. Each concept $c \in [C]$ is defined by the user through a set of $N^c \in \mathbb{N}^*$ positive examples $\mathcal{P}^c = \{\boldsymbol{x}^{c,n} \mid n \in [N^c]\}$ and a set of $N^{\neg c}$ negative examples $\mathcal{N}^c = \{\boldsymbol{x}^{\neg c,n} \mid n \in [N^{\neg c}]\}$. The concept $c \in [C]$ is present in the positive examples $\mathcal{P}^c$ and absent from the negative examples $\mathcal{N}^c$. For instance, if we work in a computer vision context and the concept of interest is $c = \text{Stripes}$, then $\mathcal{P}^c$ contains images with stripes (e.g. zebra images) and $\mathcal{N}^c$ contains images without stripes (e.g. cow images). We note that non-binary concepts (e.g. a colour) can be one-hot encoded to fit this formulation. In this paper, we use balanced sets: $N^c = N^{\neg c}$.

**CAV formalism.** Let us start by summarizing the CAV formalism [31]. This formalism relies on the crucial assumption that the sets $\boldsymbol{g}(\mathcal{P}^c) \subset \mathcal{H}$ and $\boldsymbol{g}(\mathcal{N}^c) \subset \mathcal{H}$ are linearly separable in $\mathcal{H}$. Hence, it is possible to find a vector $\boldsymbol{w}^c \in \mathbb{R}^{d_H}$ and a bias term $b^c \in \mathbb{R}$ such that $(\boldsymbol{w}^c)^\intercal \boldsymbol{h} + b^c > 0$ for all $\boldsymbol{h} \in \boldsymbol{g}(\mathcal{P}^c)$ and, conversely, $(\boldsymbol{w}^c)^\intercal \boldsymbol{h} + b^c < 0$ for all $\boldsymbol{h} \in \boldsymbol{g}(\mathcal{N}^c)$. The vector $\boldsymbol{w}^c$ is called the *concept activation vector* (abbreviated CAV) associated to the concept $c \in [C]$. Geometrically, this vector is normal to the hyperplane separating $\boldsymbol{g}(\mathcal{P}^c)$ from $\boldsymbol{g}(\mathcal{N}^c)$ in the latent space $\mathcal{H}$. Intuitively, this vector points in a direction of the latent space where the presence of the concept $c \in [C]$ increases. When $f_k(\boldsymbol{x}) = l_k \circ \boldsymbol{g}(\boldsymbol{x})$ corresponds to the predicted probability of class $k \in [d_Y]$ for $\boldsymbol{x} \in \mathcal{X}$, this insight allows us to define a class-wise conceptual sensitivity. Indeed, the directional derivative

---

[1]Note that we use bold symbols for vectors.

[2]It is understood that there exists a dictionary between the integer concept identifier and the concept's name. In this way, if the first concept of interest is *stripes*, we can write $c = 1$ and $c = \text{Stripes}$.

$S_k^c(\boldsymbol{x}) \equiv (\boldsymbol{w}^c)^\intercal \boldsymbol{\nabla}_{\boldsymbol{h}} l_k [\boldsymbol{g}(\boldsymbol{x})]$ measures the extent to which the probability of class $k$ varies in the direction of the CAV. If the sensitivity is positive $S_k^c(\boldsymbol{x}) > 0$, this indicates that the presence of concept $c$ increases the belief of the model $\boldsymbol{f}$ that $k$ is the correct class for example $\boldsymbol{x}$. Note that CAV sensitivities $S_k^c(\boldsymbol{x})$ can be generalized to aggregate the gradients between $\boldsymbol{g}(\boldsymbol{x})$ and a baseline $\bar{\boldsymbol{h}} \in \mathcal{H}$ [32]. Given a dataset of examples split by class $\mathcal{D} = \bigsqcup_{k=1}^{d_Y} \mathcal{D}_k$, where $\mathcal{D}_k$ contains only examples of class $k \in [d_Y]$, it is possible to summarize the overall sensitivity of class $k$ to concept $c$ by computing the score $\mathrm{TCAV}_k^c = |\{\boldsymbol{x} \in \mathcal{D}_k | S_k^c(\boldsymbol{x}) > 0\}| / |\mathcal{D}|$, where $|\cdot|$ denotes the set cardinality.

**The limitations of linear separability.** The linear separability of concept negatives and positives is central in the aforementioned formalism. Indeed, the existence of a separating hyperplane in $\mathcal{H}$ is necessary to define CAVs, which in turn are required to define concept sensitivity and TCAV scores. This assumption has been criticized in the literature [39, 40]. The main criticism is the following: there is no reason to expect the representation map $\boldsymbol{g}$ to linearly separate concept positive and negatives since these concept-related labels are not known by the model. Let us consider a concrete example to stress that generic classifiers should not be expected to linearly separate concepts even if they linearly separate classes. Consider a classifier $\boldsymbol{f} = \boldsymbol{l} \circ \boldsymbol{g}$, where for each class $k \in [d_Y]$: $l_k(\boldsymbol{h}) = \mathrm{Softmax}(\boldsymbol{\alpha}_k^\intercal \boldsymbol{h} + \beta_k)$ with $\boldsymbol{\alpha}_k \in \mathbb{R}^{d_H}$ and $\beta_k \in \mathbb{R}$. This parametrization is typically realized when the representation space $\mathcal{H}$ corresponds to the penultimate layer of a neural network. For concreteness, we consider a computer vision setting with the classes $k \in \{\mathrm{Tiger}, \mathrm{Lion}, \mathrm{Cow}, \mathrm{Zebra}\}$. We note that the decision boundary in $\mathcal{H}$ for any pair of class is a hyperplane. For instance, the decision boundary between the classes lion and tiger is parametrized by $l_{\mathrm{Lion}}(\boldsymbol{h}) = l_{\mathrm{Tiger}}(\boldsymbol{h})$, which corresponds to the equation of a hyperplane in $\mathcal{H}$: $(\boldsymbol{\alpha}_{\mathrm{Lion}} - \boldsymbol{\alpha}_{\mathrm{Tiger}})^\intercal \boldsymbol{h} + (\beta_{\mathrm{Lion}} - \beta_{\mathrm{Tiger}}) = 0$. In this setting, any successful classifier $\boldsymbol{f}$ requires a representation map $\boldsymbol{g}$ that linearly separates the classes in the latent space $\mathcal{H}$. We now consider the concept $c = \mathrm{Stripes}$. In this case, we can expect examples from the classes tiger and zebra in $\mathcal{P}^{\mathrm{Stripes}}$ (as both tigers and zebras have stripes) and examples from the classes lion and cow in

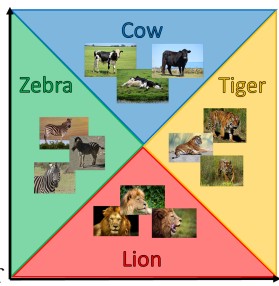

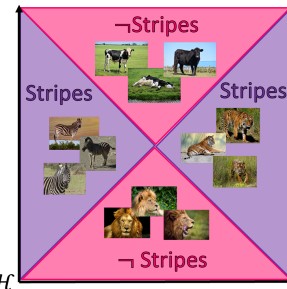

Figure 2: Classes are linearly separable (top) but concept sets are not (bottom).

$\mathcal{N}^{\mathrm{Stripes}}$ (as neither lions nor cows have stripes). As illustrated in Figure 2, it is therefore perfectly possible to have $\boldsymbol{g}(\mathcal{P}^{\mathrm{Stripes}})$ and $\boldsymbol{g}(\mathcal{N}^{\mathrm{Stripes}})$ that are not linearly separable in spite of the classes linear separability. With this example, we emphasize the subtle distinction between classes and concept linear separability. While the former is expected and holds experimentally [33], this is not the case for the latter.

**A better assumption.** Although the representations from Figure 2 do not linearly separate the concept sets, we note that concept positive and negative examples are scattered across distinct clusters. In this way, a model that produces these representations appears to make a difference between presence and absence of the concept. From this angle, we could consider that the concept is well encoded in the representation space geometry. To formalize this more general notion of concept set separability, we adapt the smoothness assumption originally formulated in semi-supervised learning [46].

**Assumption 2.1** (Concept Smoothness). A concept $c \in [C]$ is encoded in the latent space $\mathcal{H}$ if $\mathcal{H}$ is *smooth* with respect to the concept. This means that we can separate $\mathcal{H} = \mathcal{H}^c \bigsqcup \mathcal{H}^{\neg c}$ into a *concept activation region* (CAR) $\mathcal{H}^c$ where the concept $c$ is mostly present (i.e. $|\boldsymbol{g}(\mathcal{P}^c) \bigcap \mathcal{H}^c| \gg |\boldsymbol{g}(\mathcal{N}^c) \bigcap \mathcal{H}^c|$) and a region $\mathcal{H}^{\neg c}$ where the concept $c$ is mostly absent (i.e. $|\boldsymbol{g}(\mathcal{N}^c) \bigcap \mathcal{H}^{\neg c}| \gg |\boldsymbol{g}(\mathcal{P}^c) \bigcap \mathcal{H}^{\neg c}|$). If two points $\boldsymbol{h}_1, \boldsymbol{h}_2 \in \mathcal{H}$ in a high-density region of the latent space are close to each other, then we should have $\boldsymbol{h}_1, \boldsymbol{h}_2 \in \mathcal{H}^c$ or $\boldsymbol{h}_1, \boldsymbol{h}_2 \in \mathcal{H}^{\neg c}$.

*Remark* 2.1. We can easily see that linear separability is trivially included in this assumption: it corresponds to the case where the CAR $\mathcal{H}^c$ and $\mathcal{H}^{\neg c}$ are separated by a hyperplane. Conversely, our assumption does not require the CAR $\mathcal{H}^c$ and $\mathcal{H}^{\neg c}$ to be separated by a hyperplane for a concept to be relevant, as illustrated in Figure 3.

There are two crucial components in the previous assumption that need to be detailed: what do we mean by *density* and how do we extract a CAR $\mathcal{H}^c$ from $\mathcal{H}$.

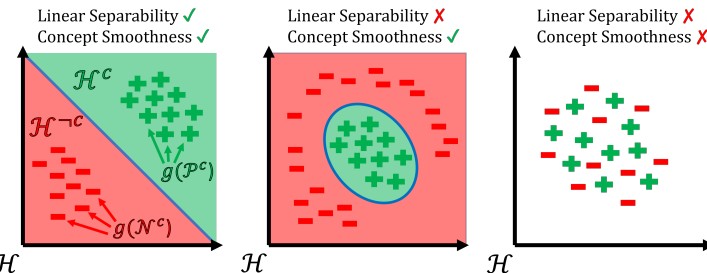

Figure 3: Linear separability implies concept smoothness but not the opposite.

## 2.2 Detecting Concepts

**Concept Density.** Let us start by detailing the notion of density that we use. We first note that the notion of proximity in $\mathcal{H}$ is crucial in Assumption 2.1. It is conventionally formalized through a kernel function $\kappa : \mathcal{H}^2 \to \mathbb{R}^+$ [47]. These functions are such that the proximity between $h_1, h_2 \in \mathcal{H}$ increases with $\kappa(h_1, h_2)$. Similarly, the proximity between $h$ and a discrete set $\mathcal{S} \subset \mathcal{H}$ of representations increases with $\sum_{h' \in \mathcal{S}} \kappa(h, h')$. This last sum can be interpreted as the density of examples from $\mathcal{S}$ at a point $h \in \mathcal{H}$. In our setup, it is natural to define a density relative to each concept $c \in [C]$ by using the two sets $\mathcal{P}^c$ and $\mathcal{N}^c$. This motivates the following definition.

**Definition 2.1** (Concept Density). Let $\kappa : \mathcal{H}^2 \to \mathbb{R}^+$ be a kernel function. For each concept $c \in [C]$, we assume that we have a positive set $\mathcal{P}^c = \{x^{c,n} \mid n \in [N^c]\}$ and a negative set $\mathcal{N}^c = \{x^{\neg c,n} \mid n \in [N^c]\}$. We define the concept density as a function $\rho^c : \mathcal{H} \to \mathbb{R}$ such that

$$\rho^c(h) = \rho^{\mathcal{P}^c}(h) - \rho^{\mathcal{N}^c}(h),$$

$$\rho^{\mathcal{P}^c}(h) = \frac{1}{N^c} \sum_{n=1}^{N^c} \kappa\left[h, g(x^{c,n})\right], \qquad \rho^{\mathcal{N}^c}(h) = \frac{1}{N^c} \sum_{n=1}^{N^c} \kappa\left[h, g(x^{\neg c,n})\right].$$

The concept density for an example $x \in \mathcal{X}$ can similarly be defined as $\rho^c[g(x)]$.

*Remark* 2.2. This density function is not necessarily positive. Indeed, we have assigned a positive contribution for examples from $\mathcal{P}^c$ and a negative one for those of $\mathcal{N}^c$. The idea is that $\rho^c(h) > 0$ whenever the density of $g(\mathcal{P}^c)$ is higher around $h$. Conversely, $\rho^c(h) < 0$ whenever the density of $g(\mathcal{N}^c)$ is higher around $h$. Finally, $\rho^c(h) \approx 0$ if $h$ is isolated from positive and negative examples or if the density of $g(\mathcal{P}^c)$ balances the density of $g(\mathcal{N}^c)$ around $h$.

**Concept Activation Regions.** With this definition, it is tempting to use the density $\rho^c$ to define the regions $\mathcal{H}^c$ and $\mathcal{H}^{\neg c}$. A natural choice would be to define the CAR as the positive density region $\mathcal{H}^c = (\rho^c)^{-1}(\mathbb{R}^+)$ and its complementary as the negative density region $\mathcal{H}^{\neg c} = (\rho^c)^{-1}(\mathbb{R}^-)$. This corresponds to using a Parzen window classifier for the concept [48]. An obvious limitation of this approach is that each evaluation of the density $\rho^c$ scales linearly with the number $N^c$ of concept examples. If the size of concept sets is large, it is possible to obtain a sparse version of this Parzen window classifier with a *support vector classifier* (SVC) [49, 50] $s_\kappa^c : \mathcal{H} \to \{0, 1\}$. This SVC is fitted to discriminate the concept sets $g(\mathcal{P}^c)$ and $g(\mathcal{N}^c)$. We can then define the CARs as $\mathcal{H}^c = (s_\kappa^c)^{-1}(1)$ and $\mathcal{H}^{\neg c} = (s_\kappa^c)^{-1}(0)$. More details can be found in Appendix A.

**Global explanations.** Our CAR formalism permits to extend those local (i.e. sample-wise) considerations globally. Let us start by defining the equivalent of TCAV scores described in Section 2.1. This score aims at understanding how the model relates classes with concepts. We define the TCAR score as the fraction of examples that have class $k$ and whose representation lies in the CAR $\mathcal{H}^c$: $\text{TCAR}_k^c = |g(\mathcal{D}_k) \cap \mathcal{H}^c| / |\mathcal{D}_k|$. Note that $\text{TCAR}_k^c \in [0, 1]$, where 0 corresponds to no overlap and 1 to a full overlap. In Appendix B, we extend this approach to measure the overlap between two concepts.

**Latent space isometries invariance.** In many applications such as clustering [51] or data visualization [52], the only relevant geometrical information of the representation space $\mathcal{H}$ is the distance $\|h_1 - h_2\|_{\mathcal{H}}$ between every pair of points $(h_1, h_2) \in \mathcal{H}^2$. Informally, we say that two representation spaces are isometric if they assign the same distance to each pair of points. In the aforementioned applications, two isometric representation spaces are therefore indistinguishable from one another.

Since concept-based explanations similarly describe the representation space geometry, one might require similar invariance to hold in this context. In Appendix D, we show that our CAR formalism provides such guarantee if $\kappa$ is a radial kernel [53]. To the best of our knowledge, this type of analysis has not been performed in the context of concept-based explanation methods. We believe that future works in this domain would greatly benefit from this type of insight.

## 2.3 Concepts and Features

Not all features are relevant to identify a concept. As an example, let us consider the concept $c = $ Stripes in a computer vision setting. If the image $\boldsymbol{x}$ represents a zebra, only some small portion of the image will exhibit stripes (namely the body of the zebra). Therefore, if we build a saliency map for the concept $c$, we expect only this part of the image to be relevant for the identification of this concept. Existing works to produce concept-level saliency maps relying on the CAV formalism exist in the literature [34, 54]. In the absence of CAVs, we need an alternative approach. We now describe how any feature importance method can be used in conjunction with CARs. A generic feature importance method assigns a score $a_i(f, \boldsymbol{x}) \in \mathbb{R}$ to each feature $i \in [d_X]$ for a *scalar* model $f : \mathcal{X} \to \mathbb{R}$ to make a prediction $f(\boldsymbol{x})$ [55]. Since our purpose is to measure the relevance of each feature in identifying a concept $c \in [C]$, we compute the feature importance for the concept density: $a_i(\rho^c \circ \boldsymbol{g}, \boldsymbol{x})$. Note that some specific feature importance methods, such as Integrated Gradients [25] or Gradient Shap [24], require the explained model to be differentiable. This is the case whenever the kernel $\kappa$ and the latent representation $\boldsymbol{g}$ are differentiable. We note that the support vector classifier $s_\kappa^c$ cannot be used in this context due to its non-differentiability. In Appendix C, we show that our CAR-based feature importance can be endowed with a useful completeness propriety that relates the importance scores $a_i(\rho^c \circ \boldsymbol{g}, \boldsymbol{x})$ to the concept density $\rho^c \circ \boldsymbol{g}(\boldsymbol{x})$.

# 3 Experiments

The code to reproduce all the experiments from this section is available at https://github.com/JonathanCrabbe/CARs and https://github.com/vanderschaarlab/CARs.

## 3.1 Empirical Evaluation

Our purpose is to empirically validate the formalism described in the previous section. We have several independent components to evaluate: ① the concept classifier used to detect the CARs $\mathcal{H}^c$, ② the global explanations induced by the TCAR values and ③ the feature importance scores induced by the concept densities $\rho^c$.

**Datasets.** We perform our experiments on 3 datasets. ① The MNIST dataset [56] consists of $28 \times 28$ grayscale images, each representing a digit. We train a convolutional neural network (CNN) with 2 layers to identify the digit of each image. ② The MIT-BIH Electrocardiogram (ECG) dataset [57, 58] consists of univariate time series with 187 time steps, each representing a heartbeat cycle. We train a CNN with 3 layers to determine whether each heartbeat is normal or abnormal. ③ The Caltech-UCSD Birds-200 (CUB) dataset [59] consists of coloured images of various sizes, each representing a bird from one of the 200 species present in the dataset. We fine-tune an Inceptionv3 neural network [60] to identify the species each bird belongs to among the 200 possible choices.

**Concepts.** For each dataset, we study the models through the lens of several well-defined concepts that are provided by human annotations. ① For MNIST: we use $C = 4$ concepts that correspond to simple geometrical attributes of the images: *Loop* (positive images include a loop), *Vertical/Horizontal Line* (positive images include a vertical/horizontal line) and *Curvature* (positive images contain segments that are not straight lines). ② For ECG: we use $C = 4$ concepts defined by cardiologists to better characterize abnormal heartbeats [61]: *Premature Ventricular*, *Supraventricular*, *Fusion Beats* and *Unknown*. The exact definition for each of these concepts is beyond the scope of this paper. We simply note that annotations for those concepts are available in the ECG dataset. ③ For CUB: we use $C = 112$ concepts that correspond to visual attributes of the birds (e.g. their size, the colour of their wings, etc.). We use the same procedure as [41] to extract those concepts from the CUB dataset. We stress that, in each case, we selected concepts that can be unambiguously associated to the classes that are predicted by the models. Hence, it is reasonable to expect those concepts to be salient for the

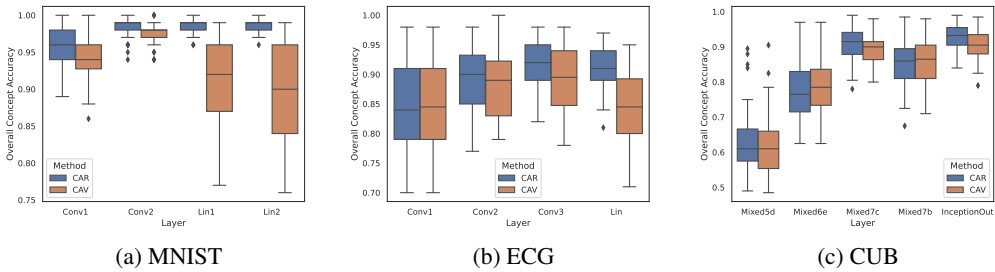

(a) MNIST $\qquad$ (b) ECG $\qquad$ (c) CUB

Figure 4: Overall accuracy of concept classifiers.

models. In each case, we sample the positive and negative sets $\mathcal{P}^c$ and $\mathcal{N}^c$ from the model's training sets. For more details on the concepts and the models, please refer to Appendix E.

### 3.1.1 Accuracy of concept activation regions

**Methodology.** The purpose of this experiment is to assess if the concept regions $\mathcal{H}^c$ identified by our CAR classifier generalize well to unseen examples. Each of the models described above are endowed with several representation spaces (one per hidden layer). For several of those latent spaces, we fit our CAR classifier (SVC with radial basis function kernel) to discriminate the concept sets $\mathcal{P}^c, \mathcal{N}^c$ for each concept $c \in [C]$. These two sets have a size $N^c = 200$ and are sampled from the model's training set. The classifier is then evaluated by computing its accuracy on a holdout balanced concept set $\mathcal{T}^c$ of size 100 sampled from the model's testing set. For MNIST and ECG, we repeat this experiment 10 times for each concept and let the sets $\mathcal{P}^c, \mathcal{N}^c, \mathcal{T}^c$ vary on each run. For comparison, we perform the same experiment with a linear CAV classifier as a benchmark. We report the overall (all the concepts together) accuracy in Figure 4.

**Analysis.** The CAR classifier substantially outperforms the CAV classifier. Note that this advantage is even more striking in the representation spaces associated to the deeper (last) DNN layers. This can be better understood through the lens of Cover's theorem [62]: the linear separation underlying CAV is usually easier to achieve in higher dimensional spaces, which corresponds to the shallower (first) DNN layers in this case. When the dimension $d_H$ of the latent space becomes comparable to the size $N^c$ of the concept sets, linear separation often fails to maintain a high accuracy. By contrast, the SVC underlying CARs manage to maintain high accuracy through the more flexible notion of concept smoothness defined in Assumption 2.1. This suggests that concepts can be well encoded in the geometry of the latent space even when accurate linear separability is not possible. We also note that the accuracy CAR classifiers seems to increase with the representation's depth. This is consistent with the behaviour of class probes [33].

**Statistical significance.** The statistical significance of the concept classifiers is evaluated with the permutation test from [32]. All of them are statistically significant with p-value $< .05$ except for some concepts classifiers that are fitted with the layers *Mixed5d* and *Mixed6e* of the CUB Inceptionv3 model. We note that those classifiers do not generalize well in Figure 4c. This suggests that deeper networks are required to identify more challenging concepts correctly.

**Take-away 1:** CAR classifiers better capture how concepts are spread across representation spaces.

### 3.1.2 Consistency of global explanations

**Methodology.** The purpose of this experiment is to assess if the aforementioned concept classifiers create meaningful global association between classes and concepts. For each model, we now focus our study on the penultimate layer. We compute the TCAV and TCAR scores for each $(\mathrm{class}, \mathrm{concept})$ pair over the whole testing set. Ideally, these scores should be correlated to the true proportion (computed with the human concept annotations) of examples from the class that exhibit the concept. To that aim, we report the Pearson correlation $r$ between each score and this true proportion in Table 1. To illustrate those global explanations, we also provide the scores for one class of each dataset in Figure 5 (for CUB, we restrict to wing colour concepts, other examples are reported in Appendix K).

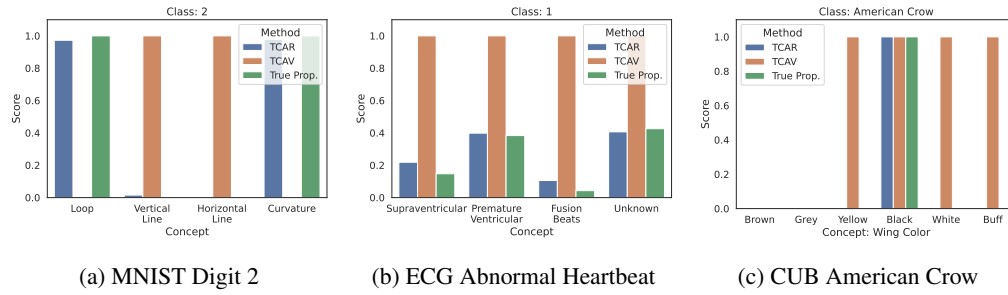

(a) MNIST Digit 2  (b) ECG Abnormal Heartbeat  (c) CUB American Crow

Figure 5: Examples of global concept-based explanations.

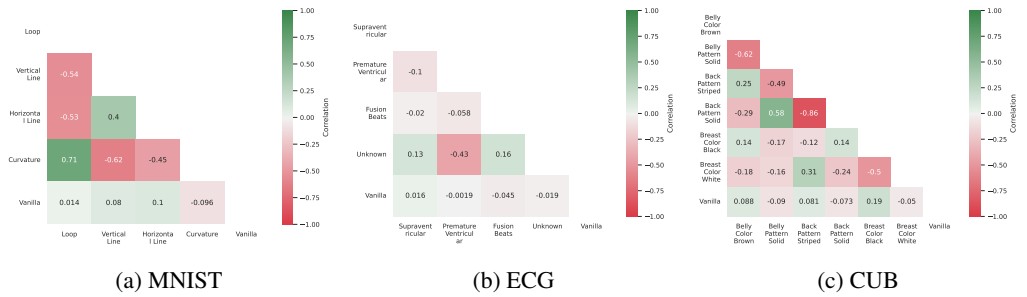

(a) MNIST  (b) ECG  (c) CUB

Figure 6: Correlations between concept feature importance.

**Analysis.** The TCAR scores better correlate with the true presence of concepts. This difference can be understood by looking at the examples from Figure 5. We note that TCAV scores tend to predict nonexistent associations (e.g. yellow wings for American crows) and miss existing associations (e.g. curvature for digit 2). For

Table 1: Evaluation of global explanations.

| Dataset | $r(\text{TCAR}, \text{TrueProp.})$ | $r(\text{TCAV}, \text{TrueProp.})$ |
|---------|------------|------------|
| MNIST | **1.00** | .60 |
| ECG | **.99** | .74 |
| CUB | **.86** | .52 |

ECG, we note that TCAR does capture the fact that some concepts (like fusion beats) are less represented within the class, while TCAV does not. In all of these cases, TCAV explanations might give the impression that the model did not learn meaningful class-concept associations. The TCAR analysis leads to the opposite conclusion. Since we have established that TCAR is built upon more accurate concept classifiers, it seems that models indeed learn concepts as intended, in spite of what TCAV explanations suggest.

**Take-away 2:** TCAR scores more faithfully reflect the true association between classes and concepts.

### 3.1.3 Coherency of concept-based feature importance

**Methodology.** Focusing again on the model's penultimate layer, we are now interested in feature importance. The evaluation of feature importance methods is a notoriously difficult problem [63]. To perform an analysis in line with our concept-based explanations, we propose to check that our concept-based feature importance meaningfully captures features that are concept-specific. We analyse this through 2 desiderata: ① concept-based feature importance is not equivalent to the model (vanilla) feature importance and ② only concepts that can be identified with similar features should yield correlated feature importance. To evaluate these desiderata empirically, we compute our CAR-based Integrated Gradients $a_i(\rho^c \circ g, x)$ for each concept $c \in [C]$ and for each example $x \in \mathcal{D}_{\text{test}}$ from the test set. Similarly, we compute the vanilla Integrated Gradients $a_i(f, x)$ for the model [3]. We quantitatively compare these various feature importance scores by computing their Pearson correlation $r$ as in [64, 65]. We report the results in Figure 6 (again, we selected a subset of 6 concepts for CUB).

---

[3]The vanilla Integrated Gradients are computed for the model estimation of the true class probability.

**Analysis.** First, we observe that CAR-based feature importance correlates weakly with vanilla feature importance ($|r| < .25$ for all datasets). This confirms that desideratum ① is fulfilled. Then, we note that most of the CAR-based feature importance scores are decorrelated or weakly correlated with each other. The counterexamples that we observe indeed correspond to concepts that can be identified with similar features. A first example is the positive correlation between the loop and the curvature concepts for MNIST ($r = .71$), both concepts are generally associated to curved symbols. Another example is the negative correlation between striped and solid back patterns for CUB birds ($r = -.86$), those concepts are mutually exclusive and identified by inspecting the bird's back. Those examples support that CAR-based feature importance fulfils desideratum ②.

**Take-away 3:** CAR-based feature importance is concept-specific and captures concept associations.

## 3.2 Use Case: Machine Learning Model Rediscovering Known Medical Concepts

We will now describe a use case of the CAR formalism introduced in this paper. We stress that the literature already contains numerous use cases of CAV concept-based explanations, especially in the medical setting [66, 67, 37]. Since CARs generalize CAVs, it goes without saying that they apply to these use cases. Rather than repeating existing usage of concept-based explanations, we discuss an alternative use case motivated by recent trends in machine learning. With the successes of deep models in various scientific domains [5–8], we witness an increasing overlap between scientific discovery and machine learning. While this new trend opens up fascinating opportunities, it comes with a set of new challenges. The evaluation of machine learning models is arguably one of the most important of these challenges. In a scientific context, the canonical machine learning approach to validate models (out-of-sample generalization) is likely to be insufficient. Beyond generalization on unseen data, the scientific validity of a model requires consistency with established scientific knowledge [68]. We propose to illustrate how our CARs can be used in this context.

**Dataset.** We use the data collected with the Surveillance, Epidemiology, and End Results (SEER) Program. The dataset [69] contains a US population-based cohort of 171,942 men diagnosed with non-metastatic prostate cancer between Jan 1, 2000, and Dec 31, 2016. Each patient is described by age, the results of a prostate-specific antigen blood test (PSA), the clinical stage of its tumour and two Gleason scores (primary and secondary). Each patient also has a label that indicates if they died because of their prostate cancer. We train a multilayer perceptron (MLP) to predict the patient's mortality on 90% of the data and test on the remaining 10%. For a more detailed description of the data and the model, please refer to Appendix F.

**Concepts.** Doctors use an established grading system to predict how likely the cancer is to spread [70]. This system assigns to each patient a grade between 1 and 5. The probability that the cancer spreads increases with this grade. It can be computed from the two Gleason scores (more details in Appendix F). We can consider each of these 5 grades as a concept. We note that this grade is not explicitly part of the input features of the MLP that we trained. Our purpose is to assess if the MLP implicitly discovered those grades in order to predict the patient's mortality. If this happens to be the case, this would demonstrate that the MLP is in-line with existing medical knowledge.

**Methodology.** As in the previous section, we are going to use the output of the MLP's penultimate layer as a representation space. We fit a CAR classifier (SVC with linear kernel) to discriminate the concepts sets $\mathcal{P}^c, \mathcal{N}^c$ for each grade $c \in [5]$. Both of those sets contain $N^c = 250$ patients sampled from the training set. We then proceed to several verifications. ① We determine if the grades have been learned by measuring the accuracy of each CAR classifier on a holdout balanced concept set $\mathcal{T}^c$ of size 100 sampled from the model's testing set. ② We check that each grade is associated to the appropriate outcome by reporting the TCAR scores between each pair $(\mathrm{mortality}, \mathrm{grade})$ in Figure 7a. ③ We verify that grades are identified with the right features by plotting a summary of the absolute feature importance scores $|a_i(\rho^c \circ \boldsymbol{g}, \boldsymbol{x})|$ on the test set $\boldsymbol{x} \in \mathcal{D}_{\mathrm{test}}$ in Figure 7b.

**Analysis.** Let us summarize the findings for each of the above points. ① All of the CAR classifiers generalize well on the test set (their accuracy ranges from 90% to 100%). This strongly suggests that the MLP implicitly separates patients with different grades in representation space. ② Figure 7a demonstrates that the model associates higher grades with higher mortality. This is in line with the clinical interpretation of the grades. ③ Figure 7b suggests that the Gleason scores constitute the most important features overall (highest quantiles) for the model to discriminate between grades. This is consistent with the fact that grades are computed with the Gleason scores. We conclude that the MLP

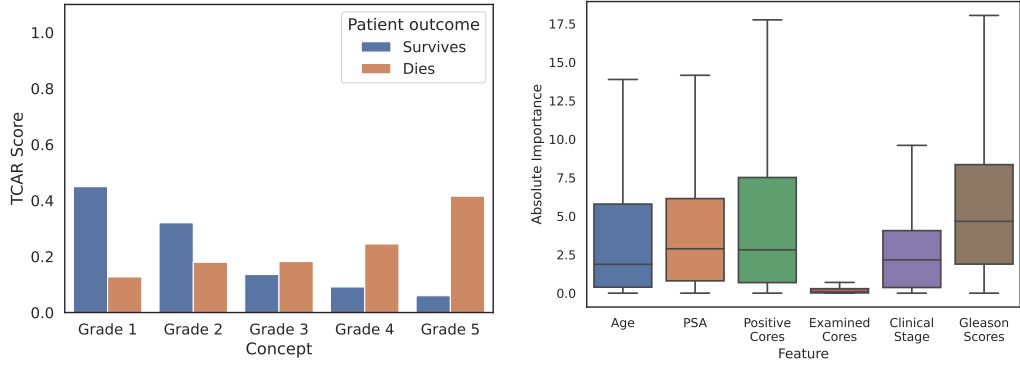

(a) TCAR grades vs mortality.

(b) Grades feature importance.

Figure 7: CAR explanations with the prostate cancer grades as concepts.

implicitly identifies the patient's grades with high accuracy and in a way that is consistent with the medical literature.

**Take-away 4:** The CAR formalism can reliably support scientific evaluation of a model.

## 4 Conclusion

We introduced Concept Activation Regions, a new framework to relax the linear separability assumption underlying the Concept Activation Vector formalism. We showed that our framework guarantees crucial properties, such as invariance with respect to latent symmetries. Through extensive validation on several datasets, we verified that ① Concept Activation Regions better capture the distribution of concepts across the model's representation space, ② The resulting global explanations are more consistent with human annotations and ③ Concept Activation Regions permit to define concept-specific feature importance that is consistent with human intuition. Finally, through a use case involving prostate cancer data, we show the neural network can implicitly rediscover known scientific concepts, such as the prostate cancer grading system.

Many important points that were not covered in the main paper can be found in the appendices. In Appendix A, we discuss how to tune the various hyperparameters of the CAR classifiers. In Appendix B, we explain how to generalize concept activation vectors to nonlinear decision boundaries. In Appendix G, we show that the CAR explanations are robust to adversarial perturbations and background shifts. In Appendix H, we demonstrate that CAR explanations can be used to relate abstract concepts discovered by self-explaining neural networks with human concepts. Finally, Appendix I illustrates how CAR explanations allow us to probe language models.

We believe that our extended concept explainability framework opens up many interesting avenues for future work. A first one would be to probe state of the art neural networks with an approach similar to Section 3. In particular, it would be interesting to analyse if improving model performance is associated with a better encoding of human concepts. A second one would be to analyze how concept discovery [38] can benefit from our generalized notion of concept activation. A more fine-grained characterization of a model's latent space is likely to improve the surfaced concept. A third one, as suggested by Section 3.2, would be to use concept-based explanations to make a better scientific assessment of neural networks. Indeed, consistency with well-established knowledge is crucial for a scientific model to be accepted.

## Acknowledgments and Disclosure of Funding

The authors are grateful to Fergus Imrie, Yangming Li and the 3 anonymous NeurIPS reviewers for their useful comments on an earlier version of the manuscript. Jonathan Crabbé is funded by Aviva and Mihaela van der Schaar by the Office of Naval Research (ONR), NSF 172251.

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
