# OpenReview forum: "Concept Activation Regions: A Generalized Framework For Concept-Based Explanations"
_NeurIPS.cc/2022/Conference — NeurIPS 2022 Accept_

### Official Review · Reviewer_qiiw · 2022-07-10

**Rating:** 7
**Confidence:** 4
**Soundness:** 3 good
**Presentation:** 3 good
**Contribution:** 3 good

**Summary:**

Recently, there has been a lot of focus on opening up the black-box DNNs via explanations. One such technique is Concept Activation Vectors (CAV), which explains the predictions of DNNs through user-specified concepts. One main shortcoming of CAV is that it assumes that examples corresponding to a concept are all mapped in a fixed direction in the DNNs latent feature space, which can be restrictive in practice. In this paper, the authors propose  Concept Activation Regions (CAR) based on kernel trick and support vector classification that relaxes this assumption.

**Questions:**

- Would it be possible to replicate the interpretations CAV offers like concept senstivity? An alternate would be to perform test time interventions on the Kernel-based Concept classifier.
- Is it possible to run CAR with concepts (or latent variables) generated by Autoencoders or VAE that are inherently noisy or abstract? Could CAR be used to analyze the concepts generated by techniques like SENN?
- How robust are explanations generated by CAR? Is it robust to change in backgrounds of the images?
- Would CAR work for other domains like NLP?
- Is it possible to relax the assumption that such a class of techniques requires additional annotation of concepts? As of now, CAR requires a user to specify the positive and negative examples for each concept.

Given that CAR assumes to have access to the feature extractor of the model, it isn't truly a black-box setup, unlike paper portrays.

**Limitations:**

I would encourage the authors to list the limitations of the proposed approach.

**Strengths And Weaknesses:**

Strengths:
- The proposed technique relaxes a fundamental assumption made in CAV, thereby increasing its effectiveness.
- The explanation generated by the proposed method is invariant under latent space isometrics.

Weaknesses:
- One of the critical weaknesses of CAR is that since explanations are generated through a Kernel-based technique, it loses the nice interpretations CAV offers, like if one increases the presence of a concept, how does it affect the model predictions.

---

> ### Author Response · Authors · 2022-08-01
> **Reviewer qiiw [3/3]**
>
> ## 4. CAR for NLP
>
> CAR is a general framework and can be used in a wide variety of domains that involve neural networks. In our paper, we show that CAR provides explanations for various modalities:
>
> 1. Large image dataset
> 2. Medical time series
> 3. Medical tabular data.
>
> As suggested by the reviewer, we perform a small experiment to assess if those conclusions extend to the NLP setting. We train a small CNN on the IMDB Review dataset to predict whether a review is positive or negative. We use Glove to turn the word tokens into embeddings. We would like to assess whether the concept $c = \mathrm{Positive \ Adjective}$ is encoded in the model's representations.
> Examples that exhibit the concept $c$ are sentences containing positive adjectives. We collect a positive set $\mathcal{P}^c$ of $N^c = 90$ such sentences. The negative set $\mathcal{N}^c$ is made of $N^c$ sentences randomly sampled from the Gutenberg Poem Dataset. We verified that the sentences from $\mathcal{N}^c$ did not contain positive adjectives. We then fit a CAR classifier on the representations obtained in the penultimate layer of the CNN.
>
>  We assess the generalization performance of the CAR classifier on a holdout concept set made of $N^c = 30$ concept positive and negative sentences (60 sentences in total). The CAR classifier has an accuracy of $87 \%$ on this holdout dataset. This suggests that the concept $c$ is smoothly encoded in the model's representation space, which is consistent with the importance of positive adjectives to identify positive reviews. We deduce that our CAR formalism can be used in a NLP setting. We believe that using CARs to analyze large-scale language model would be an interesting study that we leave for future work.
>
>
> ## 5. Using Concept Explanations without Human Annotations
>
>
> Although concept discovery without human intervention is a very interesting area, it is not the focus of our paper. However, we would like to point out that recent works have proposed to relax the necessity for human concept annotation by extracting the concepts from the model's representation space directly. For instance, in *Ghorbani, A., Wexler, J., & Kim, B. (2019). Automating Interpretability: Discovering and Testing Visual Concepts Learned by Neural Networks.*, the authors automatically extract visual concepts in the form of image segmentations. We note that the extraction happens without any human annotation. Once the concepts are identified, it is perfectly possible to use TCAR/TCAV to obtain global explanations in terms of the discovered concepts.
>
>
> ## 6. Minor Remarks
>
>
> We thank the reviewer for these additional remarks. We will make sure that to implement those changes in the manuscript.

---

> ### Author Response · Authors · 2022-08-01
> **Reviewer qiiw [2/3]**
>
> ## 3. Robustness of CAR Explanations
>
> As suggested by the reviewer, we perform an experiment to evaluate the robustness
> of CAR explanations. We start with adversarial perturbations.
> In this experiment, we work with the MNIST dataset in the same setting as
> the experiment from Section 3.1.2 from our paper. We train a CAR concept classifier
> for each MNIST concept $c \in [C]$. We use the CAR classifier to output TCAR scores
> relating the concept $c$ with each class $k \in [d_Y]$. As in the main paper, since the ground-truth association between concepts and classes is known (e.g. the class corresponding
> to digit 8 will always have the concept loop), we can compute the correlation $r(\mathrm{TCAR}, \mathrm{TrueProp})$ between
> our TCAR score and the ground-truth proportion of examples that exhibit the concept.
> In this experiment, this correlation is evaluated on a test set $\mathcal{D}\_{\mathrm{test}} = \mathcal{D}\_{\mathrm{adv}} \ \sqcup \mathcal{D}\_{\mathrm{orig}}$ that contains adversarial
> test examples $\mathcal{D}\_{\mathrm{adv}}$ and original test examples $\mathcal{D}\_{\mathrm{orig}}$. Each adversarial MNIST image $x\_{\mathrm{adv}} \in \mathcal{D}\_{\mathrm{adv}}$ is constructed by finding a small (w.r.t. the $\| \cdot \|_{\infty}$ norm) perturbation $\epsilon \in \mathbb{R}^{d_X}$ around an original test image $x \in \mathcal{X}$ that maximizes the prediction shift for the black-box $f : \mathcal{X} \rightarrow \mathcal{Y}$:
>
> $$\epsilon = \arg \max_{\tilde{\epsilon} \in \mathbb{R}^{d_X}} \mathrm{Cross Entropy}[f(x), f(x + \tilde{\epsilon})] \ s.t. \ \| \tilde{\epsilon} \|_{\infty} < .1$$
>
> The adversarial image is then defined as $x_{\mathrm{adv}} \equiv x + \epsilon$. We measure the correlation $r(\mathrm{TCAR}, \mathrm{TrueProp})$ by varying the proportion $\frac{|\mathcal{D}\_{\mathrm{adv}}|}{|\mathcal{D}_{\mathrm{test}}|}$ of adversarial examples in the test set. The results are reported bellow:
>
> |   Adversarial % |$r(\mathrm{TCAR}, \mathrm{TrueProp})$|
> |----------------:|---------:|
> |               0 | .99 |
> |               5 | .99 |
> |              10 | .99 |
> |              20 | .99 |
> |              50 | .97 |
> |              70 | .96 |
> |             100 | .92 |
>
> We observe that the TCAR scores keep a high correlation with the true proportion of examples that exhibit the concept even when all the test examples are adversarially perturbed. We conclude that TCAR explanations are robust with respect to adversarial perturbations in this setting.
>
> For completeness, we have also adapted the background shift robustness experiment in Section 7 from *Koh, P. et al. (2020). Concept Bottleneck Models*. As in our paper, we use CAR to explain the predictions of our Inception-V3 model trained on the original CUB training set. The explanations are made on test images where the background has been replaced. As Koh et al., we use the segmentation of the CUB dataset to isolate the bird on each image. The rest of the image is replaced by a random background sampled from the *Place365* dataset. This results in a test set $\mathcal{D}_{\mathrm{test}}$ with a background shift with respect to the training set. By following the approach from Section 3.1.2 of our paper, we measure the correlation $r(\mathrm{TCAR}, \mathrm{TrueProp})$ between the TCAR score and the true proportion of examples in the class that exhibit the concept for each $(\mathrm{class}, \mathrm{concept})$ pair. We measured a correlation of $r(\mathrm{TCAR}, \mathrm{TrueProp}) = .82$ in the background-shifted test set. This is close to the correlation for the original test set reported in the main paper, which suggests that CAR explanations are robust with respect to background shifts. Note that this correlation is still better than the one obtained with TCAV on the original test set.

---

> ### Author Response · Authors · 2022-08-01
> **Reviewer qiiw [1/3]**
>
> We would like to thank the reviewer for taking the time to make encouraging comments and constructive criticisms. By following the reviewer's suggestions, we were able to:
>
> 1. Generalize the TCAV sensitivity metric with our CAR formalism.
> 2. Demonstrate the utility of CAR explanations to understand abstract concepts discovered in an unsupervised fashion.
> 3. Demonstrate that our explanations are robust with respect to adversarial perturbations and background shifts.
> 4. Demonstrate that CAR classifiers can be used with NLP models.
> 5. Clarify the existing contributions for discovering concepts without human intervention.
>
> We believe that all of these points make a great addition to the manuscript.
>
> ## 1. Generalizing CAV Sensitivity Interpretations
>
> We thank the reviewer for suggesting this interesting extension. In our formalism, it is perfectly possible to define a *local* concept activation vector through the concept density $\rho^c : \mathcal{H} \rightarrow \mathbb{R}^+$ defined in Definition 2.1 from the main paper. Indeed, the vector $\nabla_{h}\rho^c[h] \in \mathcal{H}$ points in the direction of the representation space $\mathcal{H}$ where the concept density (and hence the presence of the concept) increases. Hence, this vector can be interpreted as a *local* concept activation vector. Note that this vector becomes global whenever we parametrize the concept density $\rho^c$ with a linear kernel $\kappa(h_1, h_2) = h_1^{\intercal} h_2$. Equipped with this generalized notion of concept activation vector, we can also generalize the CAV concept sensitivity $S^c_k$ by replacing the CAV $w^c$ by $\nabla_{h}\rho^c[h]$ for the representation $h = g(x)$ of the input $x \in \mathcal{X}$:
>
>
> $$S^c_k(x) \equiv (\nabla_{h} \rho^c [g(x)])^{\intercal} (\nabla_{h} l_k [g(x)]).$$
>
> In this way, all the interpretation provided by the CAV formalism are also available in the CAR formalism. This discussion has been added in Appendix B of the manuscript.
>
> ## 2. Using CAR with Unsupervised Concepts
>
>
> Our CAR formalism adapts to a wide variety of neural network architectures. As suggested by the reviewer, we use CAR to analyze the concepts discovered by a self explaining neural network (SENN) trained on the MNIST dataset. As in *Alvarez-Melis, D., & Jaakkola, T. (2018). Towards Robust Interpretability with Self-Explaining Neural Networks.*, we use a SENN of the form
>
> $$f(x) = \sum_{s=1}^S \theta_s (x) \cdot g_s(x),$$
>
> Where $h_s(x)$ and $\theta_s(x)$ are respectively the activation and the relevance of the synthetic concept $s \in [S]$ discovered by the SENN model. We follow the same training process as Alvarez-Melis et al. This yields a set of $S = 5$ concepts explaining the predictions made by the SENN $f : \mathcal{X} \rightarrow \mathcal{Y}$.
>
> We use our CAR formalism to study how the synthetic concepts $s \in [S]$ discovered by the SENN are related to the concepts $c \in \{ \mathrm{Loop}, \mathrm{Vertical \ Line}, \mathrm{Horizontal \ Line}, \mathrm{Curvature}  \}$ introduced in our paper. With our formalism, the relevance of a concept $c$ for a given prediction $x \mapsto f(x)$ is measured by the concept density $\rho^c \circ g (x)$. To analyze the relationship between the SENN concept $s$ and the concept $c$, we can therefore compute the correlation of their relevance:
>
> $$r(s, c) = \mathrm{corr}\_{X \sim P_{\mathrm{empirical}}(\mathcal{D}_{\mathrm{test}})} [\theta_s(X) , \rho^c\circ(X)]. $$
>
> When this correlation increases, the concepts $s$ and $c$ tend to be relevant together more often. We report the correlation between each pair $(s, c)$ in the bellow table.
>
> |      Correlation   $r(s , c)$       |     **Loop**  |   **Vertical Line** |   **Horizontal Line** |   **Curvature** |
> |:---------------:|:----------:|:----------------:|:------------------:|:------------:|
> | **SENN Concept 1** | -0.28 |      -0.12  |         0.26  |  0.11   |
> | **SENN Concept 2** | -0.50 |       0.71  |        -0.03 | -0.69   |
> | **SENN Concept 3** | -0.47 |       0.10  |         0.71  | -0.14   |
> | **SENN Concept 4** | -0.33 |       0.02 |        -0.06 | -0.01 |
> | **SENN Concept 5** |  0.57 |      -0.0 |        -0.63  |  0.07  |
>
> We note the following:
>
> 1. SENN Concept 2 correlates well with the Vertical Line Concept.
> 2. SENN Concept 3 correlates well with the Horizontal Line Concept
> 3. SENN Concept 5 correlates well with the Loop Concept.
> 4. SENN Concepts 1 and 4 are not well covered by our concepts.
>
> The above analysis shows the potential of our CAR explanations to better understand the abstract concepts discovered by SENN models. We believe that the community would greatly benefit from the ability to perform similar analyses for other interpretable architectures, such as disentangled VAEs.

---

> ### Author Response · Authors · 2022-08-05
>
> Dear reviewer,
>
> Once again, we would like to thank you for your feedback. We hope that our rebuttal has addressed any questions or concerns you may have had about the paper. If you have any other comments or concerns, please let us know. We would be happy to do our utmost to address them during the author-reviewer discussion period,which ends this Tuesday.
>
> Best regards.

---

> > ### Comment · Reviewer_qiiw · 2022-08-06
> > **Response to Rebuttal**
> >
> > Dear Authors,
> >
> > Thanks for addressing/answering all my questions, and I am satisfied with the responses. I have updated my scores to reflect the same.

---

### Official Review · Reviewer_znZB · 2022-07-12

**Rating:** 7
**Confidence:** 4
**Soundness:** 3 good
**Presentation:** 3 good
**Contribution:** 3 good

**Summary:**

The paper generalized TCAV's linear separable assumption to an additional kernel space separable, which is simply the smoothness theoretically. The linear classifier is replaced by a kernel SVC, and the TCAV score is replaced by a counterpart TCAR score. They then identify the SVC classifier has a higher accuracy, and results in more stable concept explanations.

**Questions:**

-- In the inception experiment, CAV has a pretty close accuracy to CAR (showing that the linear separability is not a huge issue), but the evaluation of TCAV score in table 1 seems completely wrong. The failure of TCAV in MNIST and ECG is understandable since the model seems under-represented, and an additional kernel classifier would help a lot. However, the result in Inception-v3 is not convincing.

-- the evaluation of concept-based feature importance is only a sanity check, how is this useful?



**Limitations:**

weakness 1 and 2 are not fully covered, but they do cover some limitations.

**Strengths And Weaknesses:**

strength
--  the paper targets an important question and limitation of TCAV
--  the empirical evaluation is thorough

weakness
-- the introduction of kernels may lead to much more hyper-parameter choice to use in practice
-- for very complicated concepts and networks, an additional RBF kernel may be insufficient to make the data linearly separable in practice
-- the layer selection problem is not dealt with, as TCAV suggested that sometimes earlier layers may be more fruitful even if the accuracy seems lower (for more low-level concepts). In my personal experience, using the mixed-7b layer inception-V3 for TCAV usually produces a much better result than the penultimate layer. Moreover, how does one choose the layer to apply TCAR is not addressed.

note:TCAV does not work well with the penultimate layer since d c/d activation = W_c (which is independent to the instance), and thus for all instances in the same class the directional derivative to the same concept would be fixed.

---------------------------------------------------------

I see that the advantage of CAR lies also on not using the sensitivity score, which makes sense. Changing my score to 7.

---

> ### Author Response · Authors · 2022-08-01
> **Reviewer znZB [3/3]**
>
> ## 4. Significance of Feature Importance Evaluation
>
>
> We believe that our consistency checks for concept-based feature importance demonstrate two crucial and non-trivial points on various datasets:
>
> 1. **Concept-based saliency maps are not generic.** The low correlation between vanilla saliency maps and concept-based saliency maps indicates that the latter are concept-specific. This is consistent with the fact that the features that are salient to identify a concept are not necessarily the same as the ones that are salient to predict the class.
> 2. **Concept-based saliency maps are consistent with human intuition.** The correlation between the saliency maps of each pair of concept $(c_1, c_2)$ appears to be important when $c_1$ and $c_2$ can be identified through the same input features (e.g. the *loop* and the *curvature* concepts in MNIST are both identified through pixels in the curved part of the digit). This is a way to confront our concept-based saliency maps with the ground-truth human knowledge, in the same spirit as the assessment of global explanations in Section 3.1.2. We note that the former are more difficult in practice, since no concept-specific ground-truth saliency map is available for the investigated datasets. For this reason, we use the saliency maps correlations between concepts and compare them with the human associations between those concepts.

---

> ### Author Response · Authors · 2022-08-01
> **Reviewer znZB [2/3]**
>
> ## 2. Layer Selection
>
> We would like to emphasize our CAR formalism, the user is free to to choose the layer they want to interpret. In many use cases, the user might want to interpret specific layers of the neural network based on their knowledge of the architecture. In our case, we decided to select the layer for which the concept classifiers (for both CAR and CAV) generalize better to unseen examples, as measured in our experiment from Section 3.1.1 from our paper.
>
> Following the reviewer's recommendation, we decided to repeat the comparison between TCAV and TCAR from Section 3.1.2 with the layer Mixed-7b of our Inception-V3 classifier. In doing so, we measured the following correlation between the scores and the ground-truth proportion of examples within a class that exhibit the concept:
>
> $$r(\mathrm{TCAV}, \mathrm{TrueProp}) = .46 \hspace{2cm} r(\mathrm{TCAR}, \mathrm{TrueProp}) = .71$$
> For both TCAV and TCAR, these correlations are lower than those  obtained in the model's penultimate layer. In this case, it appears that the association between classes and concepts are more meaningfully encoded in the deeper layers of the neural network. We believe that the machine learning community would greatly benefit from the ability to perform this type of analysis for various architectures.
>
> ## 3. Discrepancy between CAV and TCAV Accuracy
>
>
> We would like to thank the reviewer for pointing this out. After double checking our implementation, everything seems consistent and we are confident about the results reported in our paper. We would like to emphasize that the high accuracy of a CAV concept classifier does not guarantee that TCAV score correlates well with the ground-truth association between classes and concepts. This can be understood in the following way:
>
> * A highly accurate CAV classifier occurs when the concept sets are linearly separable in the model's representation space $\mathcal{H}$. This means that the feature extractor $g : \mathcal{X} \rightarrow \mathcal{H}$ tends to linearly separate examples that exhibit the concept from the ones that don't.
> * A high correlation between the TCAV score and the ground-truth association between classes and concept occurs when the model's prediction for each class is sensitive to the presence of the appropriate concepts. This means that for each class $k \in [d_Y]$, the label map $l_k : \mathcal{H} \rightarrow [0, 1]$ is sensitive to concepts $c \in [C]$ that are truly relevant to describe this class (e.g. MNIST images of digit 9 are sensitive to the loop concept).
>
> From the above discussion, we immediately notice that the two previous situations depend on two orthogonal parts of the model: the accuracy of the CAV classifier depends on the feature extractor $g$ and the correlation of the TCAV score with ground-truth depends on the label map $l$. In that light, these two situations appear independent from each other: it is perfectly possible to have highly accurate CAV classifiers and poor TCAV scores if concepts are well separated in the model's representation space $\mathcal{H}$ but the model's predictions are not sensitive to the right concepts. We note that this is precisely what occurs in the CUB setting, as we can observe from Figures 5.c in the main paper and Figure 15 in the supplementary material. In these figures, we observe that TCAV suggests non-existent associations, such as one between the class *black crow* and the concept *yellow wing colour*.
>
> A possible explanation for the better agreement between the quality of CAR classifiers and TCAR scores is the fact that, unlike TCAV, TCAR scores are not computed by using the sensitivity metric $S^c_k$ defined in Section 2.1 of the paper. As explained in Section 2.2, we use the concept activation regions *directly* to compute TCAR scores. This implies that TCAR scores are computed in the model's representation space $\mathcal{H}$ *directly* by analyzing how different classes are scattered across the concept clusters. We believe that this different characterization might explain the gap between TCAV and TCAR scores in terms of correlation with the ground-truth. This would suggest that TCAV's sensitivity $S^c_k$ might not be the most appropriate way to detect the association between a class and a concept. We will make sure to add this discussion in the manuscript.

---

> > ### Comment · Reviewer_znZB · 2022-08-06
> > **Can you add evaluation for a TCAR version using TCAV's sensitivity measure (which I see you mentioned that this can be done)**
> >
> > as title

---

> > > ### Author Response · Authors · 2022-08-07
> > > **TCAR Sensitivity Experiment**
> > >
> > > Dear reviewer,
> > >
> > > as requested, we have performed an analysis of TCAR by using the CAR sensitivity
> > >
> > > $$ S^c_k(x) \equiv (\nabla_{h} \rho^c [g(x)])^{\intercal} (\nabla_{h} l_k [g(x)]),$$
> > >
> > > and by then computing the modified TCAR score as in Section 2.1 of our paper:
> > >
> > > $$\mathrm{TCAR-S}^c_k = \frac{| \\{ x \in \mathcal{D}_k \mid S^c_k(x) > 0 \\} |}{| \mathcal{D} |}.$$
> > >
> > > In the same setting as in the CUB Experiment from Section 3.1.2 of our paper, we compute the correlation between this modified TCAR score and the ground-truth proportion of examples within a class that exhibit the concept. We obtain a correlation of $r(\mathrm{TCAR-S}, \mathrm{TrueProp}) = .5$, which is similar to the correlation obtained with TCAV and inferior to our standard TCAR score. This support the explanation provided in Point 3 of the rebuttal:  the concept sensitivity does *not* seem to be the most appropriate way to detect the association between a class and a concept. Using the standard definition of TCAR based on the concept activation regions leads to substantially better result, hence we recommend to use this definition to generate global explanations.

---

> ### Author Response · Authors · 2022-08-01
> **Reviewer znZB [1/3]**
>
> We would like to thank the reviewer for taking the time to make encouraging comments and constructive criticisms. By following the reviewer's suggestions, we were able to:
>
> 1. Propose a principled way to tune the CAR classifiers hyperparameters.
> 2. Analyze the quality of TCAR and TCAV explanations on another layer of the Inception-V3 neural network.
> 3. Better justify the discreancy between the quality of CAV classifiers and TCAV explanations.
> 4. Clarify the utility of our experiment with concept-based feature importance.
>
> We believe that all of these points make a great addition to the manuscript.
>
> ## 1. Hyperparameter Choice
>
> Since our CAR classifiers are kernel-based, they indeed come with extra hyperparameters (e.g. the kernel
> width). We would like to emphasize that, to ensure fair comparisons with the CAV classifiers,
> none of these hyperparameters has been optimized in the experiments from Section 3.1.1 and 3.1.2.
> We have used the default hyperparameters in the Scikit-Learn implementation of support vector classifiers.
> In all our experiments, CAR classifiers substantially outperform CAV hyperparameters without having to tune the hyperparameters.
>
> In the case where the user desires a CAR classifier that generalizes as well as possible, tuning these hyperparameters might be useful. We propose to tune the kernel type, kernel width and error penalty of our CAR classifiers $s^c_{\kappa}$ for each concept $c \in [C]$ by using Bayesian optimization
> and a validation concept set:
>
> 1. Randomly sample the hyperparameters from an initial prior distribution $\theta_h \sim P_{\mathrm{prior}}$.
> 2. Split the concept sets $\mathcal{P}^c, \mathcal{N}^c$
> into training concept sets $\mathcal{P}^c\_{\mathrm{train}}, \mathcal{N}^c\_{\mathrm{train}}$ and validation concept sets $\mathcal{P}^c_{\mathrm{val}}, \mathcal{N}^c_{\mathrm{val}}$.
> 3. For the current value $\theta_h$ of the hyperparameters, fit a model $s^c_{\kappa}$ to discriminate
> the training concept sets $\mathcal{P}^c_{\mathrm{train}}, \mathcal{N}^c_{\mathrm{train}}$.
> 4. Measure the accuracy $\mathrm{ACC}\_{\mathrm{val}} = \frac{\sum_{x \in \mathcal{P}^c_{\mathrm{val}}} \boldsymbol{1}(s^c_{\kappa}\circ \ g(x)=1) \ + \
>  \sum_{x \in \mathcal{N}^c_{\mathrm{val}}} \boldsymbol{1}(s^c_{\kappa}\circ \ g(x)=0)}{|\mathcal{P}^c_{\mathrm{val}} \ \cup \ \mathcal{N}^c_{\mathrm{val}}|}$.
> 5. Update the current hyperparameters $\theta_h$ based on $\mathrm{ACC}_{\mathrm{val}}$
> using Bayesian optimization (Optuna in our case).
> 6. Repeat 3-5 for a predetermined number of trials.
>
> We applied this process to the CAR accuracy experiment (same setup as in Section 3.1.1 of the main paper) to tune the CAR classifiers for the CUB concepts. Interestingly, we noticed no improvement
> with respect to the CAR classifiers reported in the main paper: tuned and standard CAR classifier have an average accuracy of $(93 \pm .2) \%$ for the penultimate Inception layer.
> This suggests that the accuracy of CAR classifiers is not heavily dependant on hyperparameters
> in this case. That said, we believe that the above approach to tune the hyperparameters
> of CAR classifiers might be useful in other cases, hence it has been added in Appendix A of the revised manuscript.
>
> We agree that not all concepts can be captured by CAR classifiers, even after hyperparameter
> optimization (e.g. Figure 4.c from the paper, we see that CAR classifiers don't generalize well on the layer Mixed-5d). As we argue in the paper, this is a strong indication that our concept
> smoothness assumption (Assumption 2.1) is violated. This implies that concepts are
> not smoothly encoded in the geometry of the model's representation space $\mathcal{H}$. The user can therefore deduce that the concept is unlikely to be salient to interpret $\mathcal{H}$. In that sense, the inability to fit CAR classifiers that generalize well is as informative as the ability to fit CAR classifiers that generalize well. Note that this whole reasoning is made more quantitative through statistical hypothesis testing in Section 3.1.1 of our paper.

---

> ### Author Response · Authors · 2022-08-05
>
> Dear reviewer,
>
> Once again, we would like to thank you for your feedback. We hope that our rebuttal has addressed any questions or concerns you may have had about the paper. If you have any other comments or concerns, please let us know. We would be happy to do our utmost to address them during the author-reviewer discussion period, which ends this Tuesday.
>
> Best regards.

---

### Official Review · Reviewer_f9CQ · 2022-07-15

**Rating:** 7
**Confidence:** 3
**Soundness:** 3 good
**Presentation:** 3 good
**Contribution:** 2 fair

**Summary:**

The paper proposes a novel concept-based explanation for neural networks. It builds on previously proposed formalism of concept activation vectors. This formalism is built on the idea that the features related to inputs containing a given concept and the features related to inputs that don't are separated by a hyperplane. The main contribution of the paper is to relax the latter: The core idea is that a neural nework latent space can be divided into non-linearly separated clusters. Such a network appears to encode well the presence or absence of the related concepts. The authors also adapt the CAV importance score (TCAV) to the new formalism, and show that these concepts are invariant to latent space isometries if they are based on support vector classifiers with a radial kernel. The authors then show several desirable properties for these explanations, including supporting  scientific evaluation of models. The experimental validation is based on multiple datasets from different domains.

**Questions:**

* Would it be possible to get inspiration from the insights in the paper to improve neural network training, with the purpose of increasing explainability?

* How are the proposed explanations sensitive to adversarial attacks?

**Limitations:**

As indicated in the previous section, it would be interesting to add to the paper an analysis of the sensitivity of the proposed approach to adversarial attacks.

**Strengths And Weaknesses:**

Strengths:
* The paper is clearly structured and well written.
* The relaxation of the linear separability in the latent space is sound, and the implemented concept activation regions seems to capture better the spread of concept-related features in the latent space.
* The experimental evaluation is well designed and extensive, and shows many desirable properties of the proposed approach.

Weaknesses:
* The paper considers mostly convolutional networks. It lacks analysis of more state-of-the art architectures like residual networks or transformers.

---

> ### Author Response · Authors · 2022-08-01
> **Reviewer f9CQ [2/2]**
>
> ## 3. Sensitivity to Adversarial Attacks
>
> As suggested by the reviewer, we perform an experiment to evaluate the robustness
> of CAR explanations with respect to adversarial perturbations.
> In this experiment, we work with the MNIST dataset in the same setting as
> the experiment from Section 3.1.2 from our paper. We train a CAR concept classifier
> for each MNIST concept $c \in [C]$. We use the CAR classifier to output TCAR scores
> relating the concept $c$ with each class $k \in [d_Y]$. As in the main paper, since the ground-truth association between concepts and classes is known (e.g. the class corresponding
> to digit 8 will always have the concept loop), we can compute the correlation $r(\mathrm{TCAR}, \mathrm{TrueProp})$ between
> our TCAR score and the ground-truth proportion of examples that exhibit the concept.
> In this experiment, this correlation is evaluated on a test set $\mathcal{D}\_{\mathrm{test}} = \mathcal{D}\_{\mathrm{adv}} \ \sqcup \mathcal{D}\_{\mathrm{orig}}$  that contains adversarial
> test examples $\mathcal{D}\_{\mathrm{adv}}$ and original test examples $\mathcal{D}\_{\mathrm{orig}}$. Each adversarial MNIST image $x\_{\mathrm{adv}} \in \mathcal{D}\_{\mathrm{adv}}$ is constructed by finding a small (w.r.t. the $\| \cdot \|_{\infty}$ norm) perturbation $\epsilon \in \mathbb{R}^{d_X}$ around an original test image $x \in \mathcal{X}$ that maximizes the prediction shift for the black-box $f : \mathcal{X} \rightarrow \mathcal{Y}$:
>
> $$\epsilon = \arg \max_{\tilde{\epsilon} \in \mathbb{R}^{d_X}} \mathrm{Cross Entropy}[f(x), f(x + \tilde{\epsilon})] \ s.t. \ \| \tilde{\epsilon} \|_{\infty} < .1$$
>
> The adversarial image is then defined as $x_{\mathrm{adv}} \equiv x + \epsilon$. We measure the correlation $r(\mathrm{TCAR}, \mathrm{TrueProp})$ by varying the proportion $\frac{|\mathcal{D}\_{\mathrm{adv}}|}{|\mathcal{D}\_{\mathrm{test}}|}$ of adversarial examples in the test set. The results are reported bellow:
>
> |   Adversarial % |$r(\mathrm{TCAR}, \mathrm{TrueProp})$|
> |----------------:|---------:|
> |               0 | .99 |
> |               5 | .99 |
> |              10 | .99 |
> |              20 | .99 |
> |              50 | .97 |
> |              70 | .96 |
> |             100 | .92 |
>
> We observe that the TCAR scores keep a high correlation with the true proportion of examples that exhibit the concept even when all the test examples are adversarially perturbed. We conclude that TCAR explanations are robust with respect to adversarial perturbations in this setting.
>
> For completeness, we have also adapted the background shift robustness experiment in Section 7 from *Koh, P. et al. (2020). Concept Bottleneck Models*. As in our paper, we use CAR to explain the predictions of our Inception-V3 model trained on the original CUB training set. The explanations are made on test images where the background has been replaced. As Koh et al., we use the segmentation of the CUB dataset to isolate the bird on each image. The rest of the image is replaced by a random background sampled from the *Place365* dataset. This results in a test set $\mathcal{D}_{\mathrm{test}}$ with a background shift with respect to the training set. By following the approach from Section 3.1.2 of our paper, we measure the correlation $r(\mathrm{TCAR}, \mathrm{TrueProp})$ between the TCAR score and the true proportion of examples in the class that exhibit the concept for each $(\mathrm{class}, \mathrm{concept})$ pair. We measured a correlation of $r(\mathrm{TCAR}, \mathrm{TrueProp}) = .82$ in the background-shifted test set. This is close to the correlation for the original test set reported in the main paper, which suggests that CAR explanations are robust with respect to background shifts. Note that this correlation is still better than the one obtained with TCAV on the original test set.

---

> ### Author Response · Authors · 2022-08-01
> **Reviewer f9CQ [1/2]**
>
> We would like to thank the reviewer for taking the time to make encouraging comments and constructive criticisms. By following the reviewer's suggestions, we were able to:
>
> 1. Extend our analysis beyond the architectures considered in the main paper.
> 2. Propose a new model training process to incorporate our insights on concept-based explainability.
> 3. Demonstrate that our explanations are robust with respect to adversarial perturbations and background shifts.
>
> We believe that all of these points make a great addition to the manuscript.
>
> ## 1. Analysis with Alternative Architectures
>
>
> As suggested by the reviewer, we extended our analysis to a ResNet-50 architecture.
> We fine-tuned the ResNet model on the CUB dataset and reproduced the experiment from Section 3.1.1 of our paper with this new architecture. In particular, we fit a CAR and a CAV classifier on the penultimate layer of the ResNet. We report the accuracy averaged  over the $C = 112$
> CUB concepts bellow:
>
> | ResNet Layer   |   CAR Accuracy  (mean $\pm$ sem) |   CAV Accuracy (mean $\pm$ sem) |
> |:--------|:------------------:|:------------------:|
> | Layer4  |          .89 $\pm$ .01  |          .87 $\pm$ .01  |
>
> As we can see, CAR classifiers are highly accurate to identify concepts in the penultimate ResNet layer. As in our paper, we observe that CAR classifiers outperform CAV classifiers, although the gap is smaller than for the Inception-V3 neural network. We deduce that our CAR formalism extends beyond the architectures explored in the paper and we hope that CAR will become widely used to interpret anby more architectures.
>
>
> ## 2. Increasing Explainability at Training Time
>
>
> Improving neural networks explainability at training time constitutes a very interesting area of research but is beyond the scope of our paper. That said, we believe that our paper indeed contains insights that might be the seed of future developments in neural network training. As an illustration, we consider an important insight from our paper: the fact that the accuracy of concept classifiers seems to increase with the depth of the layer for which we fit a classifier. In our paper, this is mainly reflected in Figure 4. This observation has a crucial consequence: it is not possible to reliably characterize the shallow layers in terms of the concepts we use.
>
> In order to improve the explainability of those shallow layers, one could leverage the recent developments in contrastive learning. The purpose of this approach would be to separate the concept set $\mathcal{P}^c$ and $\mathcal{N}^c$ in the representation space $\mathcal{H}$ corresponding to a shallow layer of the neural network. A practical way to implement this would be to follow *Chen, T. et al. (2020). A Simple Framework for Contrastive Learning of Visual Representations.* Assume that we want to separate concept positives and negatives in the representation space $\mathcal{H}$ induced by the shallow feature extractor $g : \mathcal{X} \rightarrow \mathcal{H}$. As Chen et al., one can use a projection head $p : \mathcal{H} \rightarrow \mathcal{Z}$ and enforce the separation of the concept sets through the contrastive loss
>
> $$ \mathcal{L}^c_{\mathrm{cont}} = \sum_{(x_i,x_j) \in (\mathcal{P}^c)^2} -\log \frac{\exp( \tau^{-1} \cdot\cos[p \circ g (x_i), p \circ g (x_j)])}{\sum_{x_k \in (\mathcal{P}^c \cup \mathcal{N}^c) \setminus \{ x_i \}} \exp(\tau^{-1} \cdot\cos[p \circ g (x_i), p \circ g (x_k)])},$$
>
> where $\cos(z_1, z_2) \equiv \frac{z_1^{\intercal}z_2}{\| z_1 \|_2 \cdot \| z_2 \|_2}$ and $\tau \in \mathbb{R}^+$ is a temperature parameter. The effect of this loss is to group the concept positive examples from $\mathcal{P}^c$ together and apart from the concept negatives $\mathcal{N}^c$ in the representation space $\mathcal{H}$. To the best of our knowledge, concept-based contrastive learning has not been explored in the literature. We believe that it would constitute an interesting contribution to the field based on the insights from our paper. For this reason, we added this discussion in Appendix H of the revised manuscript.

---

> ### Author Response · Authors · 2022-08-05
>
> Dear reviewer,
>
> Once again, we would like to thank you for your feedback. We hope that our rebuttal has addressed any questions or concerns you may have had about the paper. If you have any other comments or concerns, please let us know. We would be happy to do our utmost to address them during the author-reviewer discussion period,which ends this Tuesday.
>
> Best regards.

---

> > ### Comment · Reviewer_f9CQ · 2022-08-08
> > **Thanks for the rebuttal**
> >
> > I thank the authors for the extensive rebuttal.
> >
> > After reading the other reviews and the authors answers, I find that the authors addressed most of the concerns. I therefore raise my score accordingly.

---

### Meta-Review · Area_Chair_c8F1 · 2022-08-26

**Recommendation:** Accept
**Confidence:** Certain

**Metareview:**

All reviewers have found the paper as a solid contribution on a highly important topic, addressing the major shortcomings of the notable work, CAV in concept-based explainability area. On such shortcoming is that CAV assumes that examples corresponding to a concept are all mapped in a fixed direction in the DNNs latent feature space, which can be restrictive in practice. The proposed technique relaxes a fundamental assumption made in CAV, thereby increasing its effectiveness. As one main contribution, the reviewers have found the relaxation of the linear separability in the latent space sound, and the implemented concept activation regions well capturing the spread of concept-related features in the latent space. There are some concerns on the experimental analysis, that not many DNN architectures have been considered, lack of results without human annotations for concepts, and through robustness analyses. The authors have somewhat addressed these, although there is still some room for improvement. Overall, the positive aspects of the paper overweigh and I suggest acceptance of the paper.

**Award:**

No

---

### Decision · Program_Chairs · 2022-09-14

Accept